# Depressive symptoms and suicidal ideation associated with women's experience of recent economic intimate partner violence in Gauteng, South Africa: A cross-sectional study

Sunette Pienaar[1], Wiedaad Slemming[2,3], Mercilene Tanyaradzwa Machisa[1,4,5]*

**1** School of Public Health, Faculty of Health Sciences, University of Witwatersrand, Johannesburg, South Africa, **2** Division of Community Paediatrics, Department of Paediatrics and Child Health, Faculty of Health Sciences, University of the Witwatersrand, Johannesburg, South Africa, **3** Children's Institute, Department of Paediatrics and Child Health, Faculty of Health Sciences, University of Cape Town, Cape Town, South Africa, **4** South African Medical Research Council, Gender and Health Research Unit, Pretoria, South Africa, **5** University of KwaZulu-Natal, School of Nursing and Public Health, Durban, South Africa

* Mercilene.Machisa@mrc.ac.za

## Abstract

This study investigated the prevalence of economic intimate partner violence (IPV) and its association with mental health outcomes among women in Gauteng Province, South Africa. Economic IPV, involving controlling behaviours related to employment, earnings, housing, and financial support, is a critical yet understudied form of abuse with potential impacts on women's mental health. A secondary analysis was conducted using data from a representative survey of 491 women collected in 2010. Past-year economic IPV and other IPV forms were assessed through an adapted World Health Organization questionnaire. Depressive symptoms were measured by the Centre for Epidemiologic Studies Depression (CES-D) scale, and suicidal ideation was assessed for the past four weeks. Multivariate logistic regression models adjusted for socio-economic status, child abuse, partner control, other IPV types, and life trauma were employed to examine associations between economic IPV, depressive symptoms, and suicidal ideation. Results showed that 9% of women experienced past-year economic IPV, 24.2% reported depressive symptoms, and 8% had suicidal thoughts in the prior month. After adjusting for confounders, economic IPV remained significantly associated with increased odds of depressive symptoms and suicidal ideation. Younger women (18–24 years) were more likely to report suicidal thoughts than older women. These findings highlight economic IPV as a significant and often overlooked form of abuse that co-occurs with other IPV types and has strong links to adverse mental health outcomes. The study demonstrates the need for integrated approaches that combine economic empowerment and mental health support within gender-based violence services in Gauteng, South Africa. Policies should prioritize accessible, youth-friendly interventions with an emphasis on early detection

**Data availability statement:** Minimal data supporting the findings of this study is attached with the submission - Supporting information file S1.

**Funding:** MTM received financial and institutional support provided by the South African Medical Research Council Gender and Health Research Unit. WS received financial and institutional support provided by the Children's Institute, University of Cape Town. The funders had no role in study design, data collection and analysis, decision to publish, or preparation of the manuscript.

**Competing interests:** The authors have declared that no competing interests exist.

and support for vulnerable young women, including those out of school. Although the cross-sectional data limits causal inference, the evidence calls for ongoing community-based programs addressing economic abuse as part of comprehensive violence prevention and mental health promotion efforts.

## Introduction

Violence against women, particularly intimate partner violence (IPV) in its various forms, is a global social and public health concern with far-reaching consequences, negatively impacting women's physical and mental health, as well as their economic participation. Notably, women who experience IPV are at heightened risk for mental health problems, including depressive symptoms and suicidal ideation and vice versa. Understanding the intricate associations between women's experiences of IPV and these critical mental health outcomes is essential for effective intervention and prevention strategies.

Intimate partner violence against women in heterosexual relationships is defined as any behaviour of a male intimate partner, current or former, within the context of marriage, cohabitation or any other formal or informal union, that causes sexual, physical, or psychological harm [1]. Globally, physical, sexual, and emotional intimate partner violence (EIPV) experienced by women in heterosexual relationships has been extensively researched, and the evidence shows a high prevalence across all socio-economic divides [1]. The WHO estimates that 33% of women in Sub-Saharan Africa experienced lifetime physical, sexual or emotional IPV and 20% were victimised in the past year [1]. In South Africa, several population-based surveys, including the Demographic and Health Surveys (SADHS), have generated different estimates of the prevalence of physical, sexual, and intimate partner violence experienced by women using various methods and sampling designs [2–4]. The 2016 SADHS reported that 26% of ever-partnered women age 18 years or older have experienced physical, sexual, or emotional violence committed by a male partner in their lifetime [4]. Similarly, a recent national survey estimated that 27.0% of women experienced physical and/or sexual violence perpetrated by male partners since the age of 15 [5].

A significant challenge in understanding the economic dimensions of IPV is the scarcity of population-level data. This limitation stems from the absence of a globally standardised measurement for economic IPV, which in turn is rooted in a lack of universally agreed-upon operational definitions adaptable across diverse contexts and settings [3]. Global actors have prioritised statistical measurement of physical, sexual, and emotional intimate partner violence and placed less priority on measuring economic IPV. This is evidenced, for example, by the United Nations' Sustainable Development Goal (SDG 5), which aims to achieve gender equality and empower all women and girls [6]. SDG5 has a target to eliminate all forms of violence against all women and girls in the public and private spheres, including trafficking and sexual and other types of exploitation [6]. Scholars have defined economic IPV as a perpetrator's intentional control over a partner's ability to acquire, utilise, and maintain

economic and financial resources [7,8]. This definition highlights that while distinct, economic IPV frequently co-occurs with physical, sexual, and emotional forms of intimate partner violence [9]. South African studies indicate a high lifetime prevalence of economic IPV among women, estimated at around 43%, with past-year prevalence ranging between 10% and 15% [2,10,11].

The detrimental mental health consequences of experiencing physical, sexual, and emotional violence are well-established and extensively documented in global literature reviews [9,12–14]. Critically, research, including evidence from South African settings [3,14,15], has confirmed the intricate intersections and bidirectional relationships between women's mental ill-health and their experiences of physical, sexual, and emotional IPV. This bidirectionality manifests in two key pathways: first, experiencing physical, sexual, or emotional IPV significantly increases the risk of developing mental health problems such as depression, anxiety, post-traumatic stress disorder (PTSD), and suicidal ideation. The trauma, fear, and loss of control inherent in abusive relationships can directly trigger and exacerbate mental health conditions. Second, pre-existing mental health challenges can also increase a woman's vulnerability to experiencing physical, sexual, or emotional IPV. For instance, conditions like depression or substance use disorders might impair a woman's ability to recognise or resist abusive behaviours, or perpetrators may intentionally abuse women with pre-existing vulnerabilities. This cyclical relationship, where violence contributes to mental ill-health, which in turn can elevate the risk of further violence, underscores the complex interplay between these issues.

While the detrimental mental health consequences of physical, sexual, and emotional IPV are well-documented globally and within South Africa, population-based research examining the prevalence and pathways to mental health impacts associated with economic IPV among South African women remains limited [11]. A recent global literature review found consistent significant associations between experiencing economic IPV and depression across six of seven cross-sectional studies [16]. Additionally, two studies within this review indicated significant associations between economic IPV and suicidal ideation, with a notable concentration of this research conducted among survivors in shelter settings [16]. To our knowledge, the 2018 study by Gibbs et al. represents a crucial initial step in the South African context, quantitatively investigating the distinct associations between economic and emotional IPV and suicidal thoughts and depressive symptoms. This study, conducted with a non-population-based sample of young women participating in an intervention in eThekwini informal settlements, revealed strong independent links between both emotional and economic IPV and increased rates of depressive symptoms and suicidal ideation [11]. However, the design of the Gibbs et al., 2018 study limits the generalisability of the findings to South African women in the general population, and it is unknown whether similar associations can be established with population-based samples.

The profound impact of economic IPV on women's mental health operates through several interconnected pathways, as documented in the literature. Firstly, the frequent co-occurrence of economic IPV with emotional, physical, and sexual violence amplifies the cumulative trauma, significantly increasing the risk of severe mental health problems, including depression and suicidal ideation. Beyond this intersection, economic IPV undermines women's mental health through several interconnected pathways [16–21]. Previous research has confirmed the independent and significant associations of economic IPV with psychological distress. The denial of financial autonomy and control may make women feel helpless and powerless, which elevates symptoms of depression. Financial dependence generates chronic stress and insecurity, contributing to anxiety and depression. Perpetrators often leverage economic control to induce social isolation, thereby exacerbating loneliness and depression. Economic abuse diminishes self-esteem and worth, contributing to depressive symptoms. Critically, it creates barriers to escape and support, engendering hopelessness and elevating suicidal ideation. The direct financial strain and hardship resulting from economic abuse further contribute to mental distress and suicidal thoughts.

To contribute to filling the gap in evidence on the mental health impacts of economic IPV with women from general population samples, we conducted a secondary analysis of data from a population-based survey that was conducted in South Africa's most populous province, and which included the simultaneous measurement of the less studied economic IPV and mental health outcomes.

## Materials and methods

### Study sites and sampling design

This study utilised data collected from a survey conducted in Gauteng, South Africa, between April and July 2010 [22]. The survey employed a multi-stage randomised sampling design and used the 2001 South African census data as the sampling frame. Firstly, 75 Primary Sampling Units (PSUs) were randomly selected from the census sampling frame used in the 2001 census. Then, 37 PSUs were randomly allocated to sample women. In each woman's PSU, 20 households were randomly selected. Lastly, in each household, one eligible woman was invited to participate in the survey.

Eligibility criteria included being 18 years or older and normally living in the selected household, defined as sleeping an average of four nights per week in the household, and being mentally competent to complete the survey questionnaire. Women who were visiting selected households or who were mentally incapacitated were excluded. The survey had an overall response rate of 75% and a response rate of 73% amongst women [22]. The final survey sample size was 511 women. This secondary analysis included only participants whose information on economic IPV, depressive symptoms and suicidal ideation was available (n = 491). Women participants whose data were missing on these key variables were excluded.

### Ethics statement

The Gauteng Gender Based Violence Indicators Study was implemented by a collaborative team from the South African Medical Research Council, the University of the Witwatersrand, and Gender Links. The primary study received ethical approval from the South African Medical Research Council Ethics Review Committee in December 2009 (Reference: EC09–012). Permission to access and utilize the data generated from the primary study was obtained from the South African Medical Research Council in October 2020. Ethical approval for this secondary analysis was granted by the Human Research Ethics Committee (Medical) at the University of the Witwatersrand (Approval reference: M201066).[22].

The study adhered to the requirements of the WHO Ethical and Safety recommendations developed for research on domestic violence against women [23]. Research assistants gave participants information about the study before obtaining their written consent to participate in the survey. Participants were assured of confidentiality. They were allocated random study identification numbers to ensure anonymity of the information they provided. Women who experienced GBV received information on accessing services (Fig 1). When women requested, the researchers gave them local referrals for support [22]. The study participants were reimbursed for their time and inconvenience whilst completing the questionnaire [22].

### Data collection

Data were collected using questionnaires, which were uploaded and completed on Personal Digital Assistants (PDA). The questionnaires were developed in English and translated into local languages used in Gauteng province, South Africa, i.e., isiZulu, Sesotho, and Afrikaans [22]. Field workers received one week of training prior to the pilot study. Twenty women from a Primary Sampling Unit (PSU), which was not part of the study, participated in the questionnaire pretesting pilot. The questionnaires were either interview-administered or self-administered by field workers of the same sex. Questionnaire completion was done in complete privacy to ensure the safety of participants [22].

### Measurement and variable recoding

**Outcome variables.** The main outcome of the study was depressive symptoms, and the secondary outcome was suicidal ideation. Depressive symptoms were measured using the 20-item Centre for Epidemiological Studies Depression Scale (CES-D) **(**Cronbach's alpha = 0.93**)** (**Table 1**). The CES-D scale is a self-report measure that measures current symptoms of depression in the general population [24]. Women were asked how often they experienced depressive

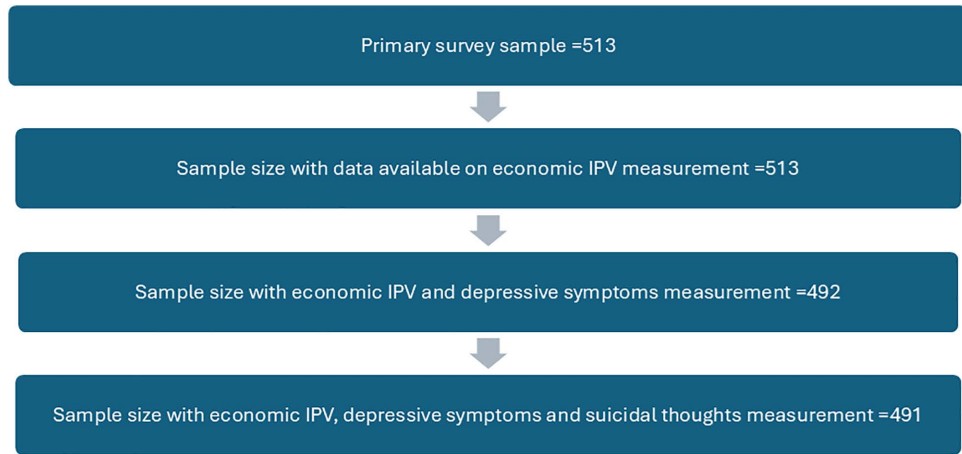

**Fig 1. Selection of the secondary study sample.**

symptoms over the past week. A 4-point Likert scale, with a range of 0 (rarely or none of the time) to 60 (more or all the time), was used to measure the frequency of depression. The scores were summed to create a binary variable. A score equal to or greater than 21 was assumed symptomatic of clinical depression, as was the case in previous studies conducted in South Africa [25–27].

Suicidal ideation was measured through a single question asking if a woman had considered ending her life in the four weeks prior to the study because of violence she experienced in her life. Responses were categorised as yes or no. This measure has been used in a previous study to investigate the adverse mental health outcomes of women in a rural area in South Africa [28].

**Exposure variables.** The exposure variables included economic, emotional, sexual, and physical IPV in the past 12 months or recent IPV by a current or former partner. The Core Questionnaire and WHO Instrument – Version 9 of the WHO Multi-country Study on Women's Health and Domestic Violence was used to ask women about their experiences of economic, emotional, sexual, and physical IPV [29]. The measurement for economic IPV included 4 items referring to being prohibited from earning an income, having earnings taken, being forced to leave the house, and having income taken. Emotional IPV was measured by six items, including controlling acts, acts that undermined women's self-esteem, or that were frightening or intimidating. Physical IPV was measured using five items referring to acts of violence such as slapping, pushing, hitting, and kicking or threats using a weapon. Sexual IPV was measured using three items of being coerced or forced to have non-consensual sex. The frequency of abuse was measured for all types of abuse along a Likert scale with four points, including options for never, once, few or many times [29].

**Confounding variables.** Socio-demographic data included education, which was categorised as no education, primary school and below, and high school and above. Also collected were age, nationality, race, income, and employment status in the year before the survey.

A modified version of the short form of the Childhood Trauma Questionnaire (CTQ) (Cronbach's alpha = 0.89) [30] (Table 1) was used to measure childhood abuse/trauma. Possible responses included never, sometimes, often, or very often. The responses for the 14 items were summed to get a child abuse score. A score lower than 14 was categorised as no child abuse, a score between 14 and 28 was considered mild child abuse experience, and a score greater than 28 as severe child abuse experience.

Relationship control was measured with the 11-item South African adaptation of the Sexual Relationship Power Scale (SRPS) (Cronbach's alpha = 0.92) (Table 1), which measures gender imbalances [31,32]. Possible responses on the

**Table 1. Definition and measurement of variables.**

| Variable | Number of items and measurement tool/description | Items |
|---|---|---|
| Outcome | | |
| Suicidal ideation | A single question was used to investigate suicidal behaviour | One question included: In the past four weeks, has the thought of ending your life been in your mind? |
| Exposure | | |
| Economic IPV in the past 12 months | Four items on experiences of economic IPV using the WHO Domestic Violence Questionnaire | Four items covering the following acts: 1.) being prohibited you from getting a job, going to work, trading, earning money or participating in income generation projects by a partner; 2.) a partner taking earnings from you; 3.) a partner forcing you or your children to leave the house where you were living and; 4.) a partner not providing money to run the house or look after the children, but has money for other things |
| Emotional IPV in the past 12 months | Four items on experiences of emotional IPV using WHO Domestic Violence Questionnaire | Emotional IPV included any of the following six acts: 1.) being insulted or made to feel bad about yourself; 2.) being belittled or humiliated in front of other people by a partner; 3.) a partner doing things to scare or intimidate on purpose for example by the way he looked, by yelling and smashing things; 4.) a partner threatening to hurt you or someone you love; 5.) a partner stopping you from seeing any of your family or friends; and 6.) a partner boasting about or bringing girlfriends home girlfriends. |
| Sexual IPV in the past 12 months | Three items including forced sex or sexual acts using WHO Domestic Violence Questionnaire | Sexual IPV items included 1.) physically forced non-consensual sex; 2.) sex because of fear of what a male partner might do; 3.) being forced to do something sexual by a male partner that they found degrading or humiliating |
| Physical IPV in the past 12 months | Five items on experiences of physical IPV using WHO Domestic Violence Questionnaire (ref) | Physical IPV items included 1.) slapping or throwing dangerous objects; 2.) pushing or shoving; 3.) hitting with a fist or other dangerous object; 4.) kicking, hitting, dragging, choking, beating, burning; 5.) threatening to use or using a gun, knife, or other weapon against you |
| Confounding | | |
| Childhood abuse/trauma | Fourteen items of the shortened Childhood Trauma Questionnaire (CTQ) (ref) | Items included: 1.) I did not have enough to eat; 2.) I lived in different households at different times; 3.) one or both parents were too drunk to take care of me; 4.) I spend time outside the home and none of the adults at home knew my whereabouts; 5.) someone touched my buttocks or genitals or made me touch them when I did not want to; 6.) I had sex with a man who was more than 5 years older than me; 7.) I had sex with someone because I was threatened or frightened or forced; 8.) I was forced to have sex against my will by a boyfriend; 9.) I was beaten at home with a belt or stick or whip or something else which was hard; 10.) I was beaten so hard at home that it left a mark or bruise; 11.) I was beaten or physically punished at school by a teacher; 12.) I saw or heard my mother being beaten by her husband or boyfriend; 13.) I was insulted or humiliated by someone in my family in front of other people; 14.) I was told I was lazy or stupid or weak by someone in my family. |
| Relationship control | Eleven items of the South African adaptation of the Sexual Relationship Power Scale (SRPS) (ref) | Items include: 1.) If I asked my partner to use a condom, he would beat or hit me; 2.) if I asked my partner to use a condom, he would get angry; 3.) my partner won't let me wear certain things; 4.) my partner has more say than I do about important decisions that affect us; 5.) my partner tells me who I can spend time with; 6.) I could leave our relationship any time I wanted to; 7.) my partner does what he wants, even if I do not want him to; 8.) when my partner and I disagree, he gets his way most of the time; 9.) because my partner buys me things, he expects me to please him; 10.) my partner always wants to know where I am; 11.) if I ask my partner to use a condom, he would think that I am having sex with other people |
| Other life trauma | Ten items of the Life Events checklist (ref) | Lifetime events items including: 1.) imprisonment/detainment, civil unrest/ war, 2.) serious injury requiring hospitalization, 3.) being close to death, 4.) witnessing a murder of family or friend, 5.) unnatural death of family or friend, 6.) witnessing the murder of stranger/s, 7.) torture, 8.) robbed or 9.) carjacked at gun or knife point and 10.) kidnapping. |

Likert scale included strongly agree = 1, agree = 2, disagree = 3, and strongly disagree = 4 to compile an SRPS score. The scores were summed to indicate different levels of relationship control. A score greater than 22 was regarded as a low score, a score between 22 and 34 was considered medium control, and scores greater than 34 indicated high control. A

lower score indicated a greater probability that the participant had less relationship power. The 10-item adapted Life Event Checklist from the PTSD Checklist (Table 1) was used to measure other life trauma [33]. The item scores were summed, with higher scores indicating greater exposure to traumatic life events.

### Data analysis

We conducted the analysis using Version 15.1 of the Stata software [34]. The study accounted for the stratification and cluster design using the (svy:) command in Stata. Bivariate analysis was conducted using cross-tabulations with measures of association. We assessed Pearson's chi-squared test to determine associations between the independent categorical variables and depressive symptoms or suicidal thoughts. Multivariate regression was used to determine the associations between past-year economic IPV experience and depressive symptoms or suicidal ideation. The analysis was adjusted for survey design to determine what proportion of women experienced IPV and depressive symptoms and suicidal ideation over the confounding variables (socio-demographic, child abuse, relationship control and other life trauma).

Factors included in the multivariable logistic regression models were those with a p-value <0.2 in the bivriate analysis for past 12-month IPV or those with theoretical relevance. Backward elimination was used to systematically remove non-significant variables and create a parsimonious model. For depressive symptoms, emotional IPV, economic IPV, sexual IPV, physical IPV, childhood trauma and other experiences of trauma were included in the regression model. The validity of the regression models for depressive symptoms and suicidal ideation was tested using the diagnostic AUC (LROC) to test whether the p-value (>0.05) has a discrimination power better than chance [35].

The Goodness-of-fit (GOF) test was used to determine whether the final models fit the data adequately (p-value>0.05), and the Linktest was used to assess the specificity of the models, whereby a p-value>0.05 indicated a well-specified model [36]. For both models, the p-value for AUC > 0.5, as well as the p-values for the GOF and Linktest, were all greater than 0.05. Statistical significance was established at the 5% level, and 95% Confidence Intervals (CI) were used to present the results. Adjusted odds ratios (aORs) were reported in the results of the multivariate logistic regression models.

## Results

### Sample description

Table 2 shows that most of the participants in the study were Black African (86.9%), South African citizens (91.7%), and were involved in an intimate heterosexual relationship in the previous year (78.8%). The participants were mostly between 35 and 49 years old (33.2%). Over three-quarters (78.6%) had a high school education; 57.4% reported receiving no employment income in the past year, and 32.2% earned R5000 or less per month. A quarter of participants (24.2%) scored above the threshold for depressive symptoms (≥21) at the time of the survey, and 7.5% reported having thoughts of suicide in the past year. Nine percent experienced economic IPV in the past 12 months.

Nine percent of women had experienced past-year economic IPV; 12.6% experienced past-year emotional IPV; 11.4% experienced past-year physical IPV, and 7.1% experienced past-year sexual IPV. Half of the participants (50.7%) reported experiencing a traumatic life event, and 95% reported experiencing childhood physical, sexual, or emotional abuse or neglect. More than a fifth of the women (21.8%) reported low sexual and decision-making power within their current relationships.

The prevalence of depressive symptoms did not differ by any of the socio-demographic characteristics (p < 0.05). The prevalence of suicidal thoughts was significantly higher among younger women ages 18–24 (35.1%) compared to older age groups. The prevalence of depressive symptoms and suicidal ideation was significantly higher among women who experienced either economic, emotional, physical, or sexual IPV compared to those who did not experience IPV. A higher proportion of those who had experienced childhood trauma reported depressive symptoms (p < 0.01) and suicidal ideation compared to those who did not experience childhood trauma (p < 0.01).

**Table 2.  Prevalence of depressive symptoms and suicidal ideation disaggregated by different participant characteristics.**

| | Total | No probable depression (score <21) | Probable depression (score ≥21) | P-value | No suicidal ideation | Suicidal ideation | P-value |
|---|---|---|---|---|---|---|---|
| | n (%) | n (%) | n (%) | | n (%) | n (%) | |
| **Socio-Demographics** | | | | | | | |
| | | | | | | | |
| **Nationality** | | | | | | | |
| South African | 450 (91.7) | 338 (90.9) | 112 (94.1) | 0.26 | 416 (91.6) | 34 (92) | 0.96 |
| Non-South African | 41(8.4) | 34 (9.1) | 7 (5.9) | | 38 (8.4) | 3 (8.1) | |
| **Race** | | | | | | | |
| Black African | 426 (86.9 | 320 (86.3) | 106 (89.1) | 0.35 | 396 (87.4) | 30 (81.1) | 0.26 |
| White | 44 (9.0) | 37 (10) | 7 (5.9) | | 38 (8.4) | 6 (16.2) | |
| Coloured, Indian or Other | 20 (4.1) | 14 (3.8) | 6 (5) | | 19 (4.2) | 1 (2.7) | |
| **Age (years)** | | | | | | | |
| 18-24years | 77 (15.7) | 61 (16.4) | 16 (13.5) | 0.81 | 64 (14.1) | 13 (35.1) | 0.01 |
| 25-34 years | 135 (27.5) | 102 (27.4) | 33 (27.7) | | 129 (28.4) | 6 (16.2) | |
| 35-49 years | 163 (33.2) | 120 (32.3) | 43 (36.1) | | 154 (33.9) | 9 (24.3) | |
| 50 + years | 116 (23.6) | 89 (23.9) | 27 (22.7) | | 107 (23.6) | 9 (24.3) | |
| **Have a husband or boyfriend (n = 486)** | | | | | | | |
| Yes | 383 (78.8) | 294 (80) | 89 (75.4) | 0.3 | 355 (78.9) | 28 (77.8) | 0.88 |
| No | 103 (21.2) | 74 (20) | 29 (24.6) | | 95 (21.1) | 8 (22.2) | |
| **Education** | | | | | | | |
| Primary school and below | 105 (21.4) | 80 (21.5) | 25 (21.0) | 0.9 | 98 (21.6) | 7 (18.9) | 0.7 |
| High school and above | 386 (78.6) | 292 (78.5) | 94 (79) | | 356 (78.4) | 30 (81.1) | |
| **Worked in the past 12-months** | | | | | | | |
| No | 139 (29.5) | 251 (70.3) | 82 (71.3) | 0.84 | 308 (70.8) | 25(67.6) | 0.68 |
| Yes | 333 (71.0) | 106 (29.7) | 33 (28.7) | | 127 (29.2) | 12(32.4) | |
| **Income per month** | | | | | | | |
| No income | 282 (57.4) | 212 (57) | 70 (58.8) | 0.87 | 263 (57.9) | 19 (51.4) | 0.85 |
| Low income | 158 (32.2) | 120 (32.3) | 38 (31.9) | | 145 (31.9) | 13 (35.1) | |
| Medium income | 28 (5.7) | 23 (6.2) | 5 (4.2) | | 25 (5.5) | 3 (8.1) | |
| High income | 23 (4.7) | 17 (4.6) | 6 (5) | | 21 (4.6) | 2 (5.4) | |
| | | | | | | | |
| | | | | | | | |
| **Childhood trauma** | | | | | | | |
| No child abuse (score<14) | 12 (2.4) | 9(4.2) | 3(2.5) | <0.01 | 10(2.2) | 2(5.4) | <0.02 |
| Mild child abuse (score 14–28) | 421 (85.7) | 336(90.3) | 85(71.4) | | 401(88.3) | 20(54.1) | |
| Severe child abuse (score>28) | 58 (11.8) | 27(7.3) | 31(26.1) | | 43(9.5) | 15(40.5) | |
| **Other life trauma** | | | | | | | |
| No life trauma | 242 (49.3) | 193 (51.9) | 49 (41.2) | 0.04 | 233 (49.1) | 19 (51.4) | 0.79 |
| Life trauma | 249 (50.7) | 179 (48.1) | 70 (58.8) | | 231 (50.9) | 18 (48.7) | |
| | | | | | | | |
| **Relationship control** | | | | | | | |
| Low (score<22) | 107 (21.8) | 73 (19.6) | 34 (28.6) | 0.11 | 93 (20.5) | 14 (37.8) | 0.01 |
| Medium (score 22–34) | 268 (54.6) | 210 (56.5) | 58 (48.7) | | 256 (56.4) | 12 (32.4) | |
| High (score<22) | 116 (23.6) | 89 (23.9) | 27 (22.7) | | 105 (23.1) | 11 (29.7) | |

*(Continued)*

**Table 2.** (Continued)

| | Total | No probable depression (score <21) | Probable depression (score ≥21) | P-value | No suicidal ideation | Suicidal ideation | P-value |
|---|---|---|---|---|---|---|---|
| | n (%) | n (%) | n (%) | | n (%) | n (%) | |
| **IPV experiences past 12 months** | | | | | | | |
| **Any economic IPV experience in past year** | | | | | | | |
| No experience | 447 (91.0) | 355 (95.4) | 92 (77.3) | <0.01 | 423 (93.2) | 24 (64.9) | <0.01 |
| Once or more | 44 (9.0) | 17 (4.6) | 27 (22.7) | | 31 (6.8) | 13 (35.1) | |
| **Any physical IPV experience in past year** | | | | | | | |
| No experience | 435 (88.6) | 261 (70.2) | 57 (48) | <0.01 | 410 (90.3) | 25 (67.6) | <0.01 |
| Once of more | 56 (11.4) | 111 (29.8) | 62 (52.1) | | 44 (9.7) | 12 (32.4) | |
| **Any emotional IPV experience in past year** | | | | | | | |
| No experience | 429 (87.4) | 345 (92.7) | 84 (70.6) | <0.01 | 405 (89.2) | 24 (64.9) | <0.01 |
| Once of more | 62 (12.6) | 27 (7.3) | 35 (29.4) | | 49 (10.8) | 13 (35.1) | |
| **Any sexual IPV experience in past year** | | | | | | | |
| No experience | 456 (92.9) | 355 (95.4) | 101 (84.9) | <0.01 | 430 (94.7) | 26 (70.3) | <0.01 |
| Once of more | 35 (7.1) | 17 (4.6) | 18 (15.1) | | 24 (5.3) | 11 (29.7) | |
| **Past year experience of both economic and physical IPV** | 30 (6.1) | 13(3.5) | 17(14.3) | <0.01 | 22(4.9) | 8(21.6)) | <0.01 |
| **Past year experience of both economic and emotional IPV** | 33 (6.7) | 14(3.8) | 19(16) | <0.01 | 25(5.5) | 8(21.6) | <0.01 |
| **Past year experience of both economic and sexual IPV** | 18 (3.7) | 7(1.9) | 11(9.2) | <0.01 | 10(2.2) | 8(21.6) | <0.01 |
| **Mental health outcomes** | | | | | | | |
| **Depression scale** | | | | | | | |
| No depression (<21) | 372 (75.8) | – | – | – | – | – | – |
| Probable depression (≥21) | 119 (24.2) | | | | | | |
| Mean | 14.1 (12.3) | – | – | – | – | – | – |
| Range | 1–60 | | | | | | |
| **Past 4-week suicidal ideation** | | – | – | – | – | – | – |
| No | 454 (92.5) | | | | | | |
| Yes | 37 (7.5) | | | | | | |

**Table 3** shows the results of the bivariate logistic regression analysis. We found no significant association between socio-demographic characteristics and depressive symptoms. Women who experienced any form of economic, emotional, physical, or sexual IPV in the past year were more likely to have depressive symptoms and suicidal thoughts compared to women who had not experienced IPV. Women aged 24 years and older were less likely to report suicidal thoughts compared to younger women aged 18–24 years. Women with mid-level control in their relationships were also less likely to be suicidal than those who had less control

**Table 4** shows the associations between variables from a multivariable logistic regression analysis. Women who experienced economic IPV in the past year were twice more likely to have depressive symptoms compared to women who had not experienced economic IPV in the same period and were four times more likely to experience suicidal thoughts as compared to women who did not experience economic IPV in the past 12 months. Experiencing emotional IPV in the past year was associated with depressive symptoms. Experiencing sexual IPV in the past year was also associated with

**Table 3. Factors associated with depressive symptoms and suicidal ideation from bivariate logistic regression analysis.**

| | Depressive Symptoms | | | Suicidal Ideation | | |
|---|---|---|---|---|---|---|
| | Odds Ratio (OR) | 95% CI | p-value | Odds Ratio (OR) | 95% CI | p-value |
| Socio-Demographics | | | | | | |
| Nationality | | | | | | |
| South African (ref) | 1 | | | 1 | | |
| Non-South African | 0.6 | 0.3 – 1.3 | 0.21 | 1 | 0.2 – 4.5 | 0.96 |
| Race | | | | | | |
| Black African (ref) | 1 | | | 1 | | |
| White | 0.6 | 0.1 – 2.3 | 0.42 | 2.1 | 0.8 – 5.7 | 0.15 |
| Coloured, Indian or Other | 1.3 | 0.5 – 3.5 | 0.53 | 0.7 | 0.2 – 3.1 | 0.62 |
| Age (years) | | | | | | |
| 18-24years (ref) | 1 | | | 1 | | |
| 25-34 years | 1.2 | 0.6 – 2.7 | 0.58 | 0.2 | 0.8 – 0.7 | 0.01 |
| 35-49 years | 1.4 | 0.6 – 3.1 | 0.45 | 0.3 | 0.1 – 0.7 | 0.01 |
| 50+years | 1.2 | 0.5 – 2.4 | 0.69 | 0.4 | 0.2 – 1.1 | 0.07 |
| Have a husband or boyfriend (n=486) | | | | | | |
| Yes (ref) | 1 | | | 1 | | |
| No | 0.7 | 0.4 – 1.4 | 0.38 | 0.9 | 0.4 – 2.4 | 0.89 |
| Education (n=468) | | | | | | |
| Primary school and below (ref) | 1 | | | 1 | | |
| High school and above | 1 | 0.6-1.7 | 0.9 | 1.2 | 0.5 – 3.0 | 0.7 |
| Worked in the past 12-months (n=472) | | | | | | |
| No (ref) | 1 | | | 1 | | |
| Yes | 1 | 0.7 – 1.6 | 0.82 | 0.9 | 0.5 – 1.5 | 0.6 |
| Income per month | | | | | | |
| No income (ref) | 1 | | | 1 | | |
| Low income | 1 | 0.6 – 1.5 | 0.85 | 1.2 | 0.6 – 2.8 | 0.59 |
| Medium income | 0.7 | 0.2 – 1.7 | 0.39 | 1.7 | 0.4 – 6.6 | 0.46 |
| High income | 1.1 | 0.4 – 3.0 | 0.9 | 1.3 | 0.3 – 5.5 | 0.7 |
| Childhood trauma | | | | | | |
| No child abuse (ref) | 1 | | | 1 | | |
| Child abuse | 1 | 0.2 – 4.4 | 0.95 | 0.2 | 0.7 – 1.0 | 0.04 |
| IPV experiences past month | | | | | | |
| Any economic IPV experience in past year | | | | | | |
| No experience (ref) | 1 | | | 1 | | |
| Once or more | 6.1 | 3.5 – 10.6 | <0.01 | 7.4 | 3.5 – 16.0 | <0.01 |
| Any physical IPV experience in past year | | | | | | |
| No experience (ref) | 1 | | | 1 | | |
| Once or more | 4.1 | 2.1 – 8.0 | <0.01 | 4.5 | 1.9 – 10.5 | <0.01 |
| Any emotional IPV experience in past year | | | | | | |
| No experience (ref) | 1 | | | 1 | | |
| Once or more | 5.3 | 3.3 – 8.5 | <0.01 | 4.5 | 1.9 – 10.3 | <0.01 |
| Any sexual IPV experience in past year | | | | | | |
| No experience (ref) | 1 | | | 1 | | |
| Once or more | 3.7 | 1.8 – 7.9 | <0.01 | 7.6 | 3.5 – 16.3 | <0.01 |
| Past year experience of both economic and physical IPV *(ref=no IPV experience)* | 4.6 | 2.5 – 8.4 | <0.01 | 5.4 | 2.2 – 13.2 | <0.01 |

*(Continued)*

**Table 3.** (Continued)

|  | Depressive Symptoms | | | Suicidal Ideation | | |
|---|---|---|---|---|---|---|
|  | Odds Ratio (OR) | 95% CI | p-value | Odds Ratio (OR) | 95% CI | p-value |
| Past year experience of both economic and emotional IPV *(ref = no IPV experience)* | 4.9 | 2.9 – 8.1 | <0.01 | 4.7 | 2.0 – 11.2 | <0.01 |
| Past year experience of both economic and sexual IPV (ref = no IPV experience) | 5.3 | 2.4 – 11.8 | <0.01 | 12.2 | 4.2 – 36.1 | <0.01 |

**Table 4. Multivariate logistic regression results for factors associated with depressive symptoms and suicidal ideation.**

|  | Depressive Symptoms | | | Suicidal Ideation | | |
|---|---|---|---|---|---|---|
|  | Adjusted Odds Ratio (aOR) | 95% CI | p-value | Adjusted Odds Ratio (aOR) | 95% CI | p-value |
| Socio-Demographics |  |  |  |  |  |  |
| Age (years) |  |  |  |  |  |  |
| 18-24years (ref) | – | – | – | 1 |  |  |
| 25-34 years | – | – | – | 0.1 | 0.04 – 0.6 | 0.01 |
| 35-49 years | – | – | – | 0.2 | 0.08 – 0.6 | 0 |
| 50 + years | – | – | – | 0.5 | 0.2 – 1.3 | 0.12 |
| IPV experiences past 12months |  |  |  |  |  |  |
| Physical IPV |  |  |  |  |  |  |
| No experience (ref) | 1 |  |  | 1 |  |  |
| Once of more | 1.2 | 0.5 – 3.1 | 0.66 | 0.9 | 0.2 – 5.0 | 0.92 |
|  |  |  |  |  |  |  |
| Emotional IPV |  |  |  |  |  |  |
| No experience (ref) | 1 |  |  | 1 |  |  |
| Once of more | 2.3 | 1.2 – 4.4 | 0.01 | 1.7 | 0.4 – 7.9 | 0.5 |
| Sexual IPV |  |  |  |  |  |  |
| No experience (ref) | 1 |  |  | 1 |  |  |
| Once of more | 1 | 0.5 – 3.3 | 1 | 4.1 | 1.2 – 14.0 | 0.02 |
| Economic IPV |  |  |  |  |  |  |
| No experience (ref) | 1 |  |  | 1 |  |  |
| Once of more | 2.3 | 1.2 – 5.0 | 0.02 | 3.9 | 1.2 – 12.4 | 0.02 |
| Childhood trauma |  |  |  |  |  |  |
| No child abuse (ref) | 1 |  |  |  |  |  |
| Child abuse | 0.8 | 0.2-4.0 | 0.78 | – | – | – |
| Other life trauma |  |  |  |  |  |  |
| No life trauma | 1 |  |  | – | – | – |
| Life trauma | 1.3 | 1.0 – 1.9 | 0.13 | – | – | – |

*ref is the reference category. The LROC (AUC) test had a p-vale >0.5 for the model for depressive symptoms and the p-values for the GOF and for the Linktest were >0.05 indicating that the models fit the data adequately and that the models were well specified.

having suicidal thoughts. Women ages 25–49 were less likely to have suicidal thoughts compared to younger women ages 18–24.

## Discussion

The overall aim of this study was to describe the prevalence of women's experiences of economic IPV, depressive symptoms and suicidal ideation among a representative sample of women from Gauteng, South Africa's most populous

province. Nearly a quarter of the women in this sample experienced economic IPV in their lifetime, and 9% women experienced economic IPV in the past 12 months. Twenty-two percent reported depressive symptoms, and 8% reported thoughts of suicide. This study found associations between experiences of economic IPV and depressive symptoms and suicidal thoughts: women who experienced economic IPV were twice more likely to report depressive symptoms and four times more likely to report suicidal thoughts in comparison to women who did not experience economic IPV. Emotional and sexual IPV experience was also associated with suicidal thoughts and depressive symptoms in multivariate regression analysis. Younger women had a greater likelihood of reporting suicidal thoughts.

This study builds upon a well-established body of research [3,15,14], demonstrating the significant associations between physical, sexual, and emotional intimate partner violence (IPV) and mental ill-health symptoms. Our findings extend this understanding by providing robust evidence, derived from a province-wide representative sample in South Africa, that economic IPV is also a prevalent experience for women and is significantly associated with adverse mental health outcomes. This corroborates earlier findings from intervention-based samples in KwaZulu Natal [11], suggesting that the detrimental mental health impact of economic IPV is not limited to specific populations.

Furthermore, our study offers a valuable temporal comparison. The comparable prevalence rates of economic IPV observed in our province-wide survey with those reported by Jewkes et al. [2] over a decade prior in three different South African provinces (i.e., 10.2-15%) suggest a concerning consistency in the prevalence of this form of abuse across time and geographical regions. The new knowledge contributed by this study lies in its confirmation of the enduring and geographically widespread nature of economic IPV in South Africa, alongside its significant mental health implications, using a contemporary, representative sample from a previously unexamined province. This strengthens the evidence base and underscores the urgent need for targeted interventions and policy responses to address this pervasive form of violence.

The present study corroborates existing evidence from South Africa and beyond, firmly establishing the significant association between economic IPV, depressive symptoms and suicidality. A substantial body of research, both within South Africa and globally, that identifies material deprivation or restrictions to accessing material resources as critical social determinants of depression [15,16,19,37–39]. The findings reinforce the understanding that when male partners restrict women's access to, control over, and ability to maintain economic and financial resources, this creates chronic stressors that significantly elevate their risk for depression and suicidality.

While the current study did not explicitly explore the pathways linking these experiences, prior research has elucidated mechanisms through which economic control and deprivation contribute to mental ill-health, including the creation of chronic stress, reduced autonomy and self-efficacy, and limited access to protective resources. The deliberate and controlling behaviours exhibited by perpetrators of economic IPV, which limit women's income-generating opportunities and facilitate economic dependence, directly contribute to this increased risk [21,38,40]. Consistent with previous research, our findings indirectly highlight the reduced economic self-sufficiency experienced by women facing economic IPV, a factor known to be protective against depression [38]. This limited control over economic resources, often leading to economic dependency, has also been identified as a key reason why individuals remain in abusive relationships [21]. Moreover, the controlling behaviours associated with economic IPV can extend to restricting women's social interactions, thereby limiting their access to work-based and other social networks, which are vital for mental well-being [16,19,20]. Our findings underscore the importance of considering economic IPV as a critical factor in understanding and addressing women's mental health in this context and highlight the need for further research to delineate the specific pathways at play within the South African setting.

Our findings reinforce the documented pathways between experiencing IPV, including the often-overlooked economic dimension, depressive symptoms, and suicidality among women survivors in South Africa, mirroring trends observed in broader international research. They align with a substantial body of evidence, both within South Africa and internationally, highlighting a strong positive association between economic hardship and suicidality [11,15–19]. This suggests that economic factors are significant contributors to suicidal thoughts and behaviours. It is well-established that suicidality

frequently presents with complex co-morbidities, including depression, post-traumatic stress, and anxiety [3,12,41]. Consistent with previous research in South African contexts, our study underscores that women experiencing economic IPV often endure other overlapping forms of abuse, such as sexual, physical, and emotional violence, and are consistently exposed to multiple traumatic experiences [16–18,21]. The cumulative impact of these intersecting forms of violence significantly elevates the likelihood of suicidal ideation among survivors.

The implications of this study are profound, particularly in highlighting the significant burden of economic IPV on women's mental health, comparable to other forms of IPV. Our findings underscore that depressive symptoms and suicidal ideation are prevalent across all socioeconomic strata, reflecting the pervasive nature of stressors like IPV in South Africa. Crucially, the study identifies younger women as a particularly vulnerable group, facing heightened risks of suicidal thoughts. This increased vulnerability in younger women may stem from a combination of factors, including the early onset of feelings of hopelessness and despair, and potentially, greater exposure to recent IPV experiences. While the associations between economic IPV and suicidality appear to diminish with age, this may be attributed to either reduced exposure to trauma or the development of increased resilience over time. These findings necessitate a multifaceted approach. First, interventions must target young women specifically, incorporating gender-transformative strategies to challenge inequitable norms and improve their agency. Concurrently, such interventions should prioritize women's economic empowerment, reducing dependency on abusive partners and fostering financial autonomy. Second, mental health services and support services for IPV survivors must be re-oriented towards community-based and primary healthcare settings, ensuring accessibility and responsiveness to the specific needs of women, especially young women. Third, research on economic IPV should prioritize the development and use of validated measurement tools to accurately assess its prevalence and its distinct impact on mental health outcomes. Finally, policy and legal frameworks, such as the Domestic Violence Act, require strengthened implementation and enhanced guidelines to effectively protect women from all forms of IPV, including economic abuse.

The findings of this study should be interpreted in light of certain limitations. One notable limitation is the use of survey data collected in 2010. Since this period, only a limited number of population-based surveys measuring gender-based violence (GBV) in South Africa have been conducted, with the recent completion of the first national GBV survey being a major advancement [5]. The 2010 data, drawn from the most populous of South Africa's nine provinces, provided a valuable opportunity to interrogate the research questions and generate important insights at that time. However, the ongoing analysis of the recently completed national survey will be critical to understanding how changes over the past decade—in women's economic empowerment, GBV policy frameworks, prevention programming, and service availability—have influenced the prevalence of women's experiences of economic intimate partner violence (IPV) and their associations with mental health observed in this study.

Additionally, the cross-sectional design precludes definitive conclusions regarding causality. Specifically, we were unable to establish the temporal relationship between depressive symptoms and suicidal thoughts, limiting our ability to determine whether these preceded or resulted from experiences of economic IPV. Moreover, the relatively low prevalence of past-year IPV within the sample may have limited the study's statistical power to detect associations and estimate effect sizes with greater precision. Suicidal ideation was assessed using a single item may have led to an underestimation of its true prevalence. Although we identified strong associations between past 12-month IPV and suicidal ideation, some categories exhibited wide or spurious confidence intervals, indicating reduced precision in these specific associations. Future research should address these limitations. Longitudinal or cohort studies with larger sample sizes are crucial to more definitively elucidate the nature and directionality of the association between economic IPV and mental health outcomes in South African settings.

## Conclusion

This study demonstrates the significance of acknowledging the often-overlooked economic dimensions of IPV, demonstrating its co-occurrence with other, more frequently studied forms of abuse against women in Gauteng, South Africa,

and crucially, highlighting its detrimental impact on mental health. The findings underscore the potential of sustained economic empowerment programs to enhance women's overall well-being. Coupled with this, interventions must enhance critical consciousness and agency, enabling women to challenge inequitable power dynamics and mitigate their vulnerability to economic control. Given the potential bidirectionality of the associations between IPV and mental health, comprehensive mental health support, integrated within accessible and youth-friendly GBV services, is essential for survivors.

Critically, these findings emphasize the urgency of early interventions, especially for younger women aged 18–24, identified in this study as a particularly vulnerable population facing increased risks of depression and suicidality. It is also important to note that interventions must be designed to reach out-of-school youth through community-based programs, utilizing peer educators and digital platforms to disseminate information, promote help-seeking behaviour, and facilitate access to services.

## Supporting information

**S1 Data.**
(XLSX)

## Acknowledgments

We acknowledge Gender Links, the South African Medical Research Council and the University of the Witwatersrand for conducting the primary survey—the GBV Indicator Survey in Gauteng—and for providing access to the dataset utilized in this study. Special appreciation is extended to Dr. Zvifadzo Matsena-Zingoni for their invaluable support to SP in conducting the secondary analysis and for their guidance in the use of STATA. Finally, we acknowledge the women from Gauteng Province, South Africa, who consented to participate in this important study; without their contribution, this research would not have been possible.

## Author contributions

**Conceptualization:** Sunette Pienaar, Wiedaad Slemming, Mercilene Tanyaradzwa Machisa.

**Data curation:** Mercilene Tanyaradzwa Machisa.

**Formal analysis:** Sunette Pienaar, Mercilene Tanyaradzwa Machisa.

**Funding acquisition:** Mercilene Tanyaradzwa Machisa.

**Investigation:** Mercilene Tanyaradzwa Machisa.

**Methodology:** Mercilene Tanyaradzwa Machisa.

**Project administration:** Sunette Pienaar.

**Resources:** Wiedaad Slemming, Mercilene Tanyaradzwa Machisa.

**Software:** Sunette Pienaar, Wiedaad Slemming.

**Supervision:** Wiedaad Slemming, Mercilene Tanyaradzwa Machisa.

**Validation:** Wiedaad Slemming, Mercilene Tanyaradzwa Machisa.

**Visualization:** Sunette Pienaar, Mercilene Tanyaradzwa Machisa.

**Writing – original draft:** Sunette Pienaar.

**Writing – review & editing:** Wiedaad Slemming, Mercilene Tanyaradzwa Machisa.

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
