## [Decision Letter · Decision Letter 0]

11 Sep 2025

PGPH-D-25-02434

Depressive symptoms and suicidal ideation associated with women’s experience of recent economic intimate partner violence in Gauteng, South Africa: A cross-sectional study

Dear Dr. Machisa,

Thank you for submitting your manuscript to PLOS Global Public Health. After careful consideration, we feel that it has merit but does not fully meet PLOS Global Public Health’s publication criteria as it currently stands. Therefore, we invite you to submit a revised version of the manuscript that addresses the points raised during the review process.

We look forward to receiving your revised manuscript.

Kind regards,

Dvora Joseph Davey

Academic Editor

Journal Requirements:

1. We noticed you have some minor occurrence of overlapping text with the following previous publication(s), which needs to be addressed:

https://www.emerald.com/books/edited-volume/10989/chapter-abstract/80739948/Examining-the-Promise-and-Delivery-of-Sustainable?redirectedFrom=fulltext

In your revision ensure you cite all your sources (including your own works), and quote or rephrase any duplicated text outside the methods section. Further consideration is dependent on these concerns being addressed.

i. State the initials, alongside each funding source, of each author to receive each grant.

ii. State what role the funders took in the study. If the funders had no role in your study, please state: “The funders had no role in study design, data collection and analysis, decision to publish, or preparation of the manuscript.”

3. Please ensure that your Ethics Statement is available in its entirety at the beginning of your Methods section, under a subheading 'Ethics Statement'.

4. Please upload separate figure files in .tif or .eps format. Also, remove the figures from your manuscript file but keep the legends.

5. We have noticed that you have uploaded Supporting Information files, but you have not included a list of legends. Please add a full list of legends for your Supporting Information files after the references list.

6. We note that there is identifying data in the Supporting Information file ‘Depression suicidal economic IPV minimal 20082025.xls’. Due to the inclusion of these potentially identifying data, we have removed this file from your file inventory. Prior to sharing human research participant data, authors should consult with an ethics committee to ensure data are shared in accordance with participant consent and all applicable local laws.

 -Location data

Additional Editor Comments (if provided):

Reviewer #1:

Reviewer #2:

Reviewers' comments:

Reviewer's Responses to Questions

**Comments to the Author**

1. Does this manuscript meet PLOS Global Public Health’s publication criteria?

Reviewer #1: Partly

Reviewer #2: Yes

2. Has the statistical analysis been performed appropriately and rigorously?

Reviewer #1: Yes

Reviewer #2: Yes

3. Have the authors made all data underlying the findings in their manuscript fully available (please refer to the Data Availability Statement at the start of the manuscript PDF file)?

Reviewer #1: Yes

Reviewer #2: Yes

4. Is the manuscript presented in an intelligible fashion and written in standard English?

Reviewer #1: Yes

Reviewer #2: Yes

Reviewer #1: This manuscript addresses an important but understudied topic: economic IPV and its association with depressive symptoms and suicidal ideation among women in Gauteng, South Africa. Strengths include use of population-based data, validated tools, and policy-relevant implications. However, I recommend major revision. Key concerns: (1) the dataset is from 2010 and this limitation must be emphasized; (2) suicidal ideation was measured with a single item tied to violence, which likely underestimates prevalence; (3) causal phrasing (“doubles the risk”) should be revised to reflect associations; (4) low prevalence of economic IPV limits statistical power and precision; and (5) conclusions should not overgeneralize beyond Gauteng. The introduction and discussion could be streamlined, tables simplified, and language edited. With these revisions, the study would make a valuable contribution.

Reviewer #2: The study addressed an important gap in understanding economic intimate partner violence and its mental health impacts using population-based data from South Africa. The strengths of the paper include: The secondary analysis is appropriate for exploring the research questions using existing population-based data, multi-stage randomized sampling design provided good representativeness for Gauteng Province, the use of validated instrument (CES-D for depression, WHO questionnaire for IPV) is methodologically sound, appropriate statistical approach with multivariate logistic regression controlling for relevant confounders, and good response rate (75% overall, 73% for women).

However, several areas require attention for improvement.

1. The abstract does not adhere to the journal’s guidelines. It exceeded the 300-word limit and uses subheadings. Please revise it to be a single, concise paragraph of not more than 300 words, containing the required information (background, objectives, methods, results and conclusion).

2. In the abstract section, the data collection date (the survey year, 2010) should be mentioned upfront for transparency.

3. Data from 2010 is now 15 years old – IPV patterns and mental health service availability may have changed significantly. The authors may wish to discuss limitations and current relevance

4. Age categorisation seems arbitrary (18-24, 25-34, 35-49, 50+) without justification (Table 2). The authors should either provide a rationale for age categorisation or use standard epidemiological age groups.

5. The manuscript used causal language (e.g., increased risk – line 307 and 356, heightened risk – line 384) in the results and discussion sections. However, in a cross-sectional study, it can only demonstrate associations, not causality. The authors may wish to revise the language throughout the manuscript to reflect association rather than causation. Alternative phrasing:

a. Change “A increases the risk of B” to “A is associated with B.”

b. Chane “A leads to B” to “A is linked to B.”

6. Table 2 (Line 567) shows 24.2% with probable depression, but the abstract says “twenty-four per cent” – these should match exactly. Some percentages in tables don’t align with the stated sample size

**Do you want your identity to be public for this peer review?** For information about this choice, including consent withdrawal, please see our Privacy Policy

Reviewer #1: No

Reviewer #2: No

---

## [Editor Report · Decision Letter 1]

14 Oct 2025

Depressive symptoms and suicidal ideation associated with women’s experience of recent economic intimate partner violence in Gauteng, South Africa: A cross-sectional study

PGPH-D-25-02434R1

Dear Prof Machisa,

We are pleased to inform you that your manuscript 'Depressive symptoms and suicidal ideation associated with women’s experience of recent economic intimate partner violence in Gauteng, South Africa: A cross-sectional study' has been provisionally accepted for publication in PLOS Global Public Health.

Best regards,

Dvora Joseph Davey

Academic Editor